# Design of a Two-Dimensional Conveyor Platform with Cargo Pose Recognition and Adjustment Capabilities

**DOI:** 10.3390/s23218754

**Published:** 2023-10-27

**Authors:** Zhiguo Zhou, Hui Zhang, Kai Liu, Fengying Ma, Shijie Lu, Jian Zhou, Linhan Ma

**Affiliations:** School of Information and Automation Engineering, Qilu University of Technology (Shandong Academy of Sciences), Jinan 250300, China; 13666392352@163.com (Z.Z.); liukaiyy2017@163.com (K.L.); mafengy@163.com (F.M.); 15684705112@163.com (S.L.); 18853104864@163.com (J.Z.); Mlinhan906@163.com (L.M.)

**Keywords:** information fusion, pose recognition, two-dimensional conveying

## Abstract

Linear conveyors, traditional tools for cargo transportation, have faced criticism due to their directional constraints, inability to adjust poses, and single-item conveyance, making them unsuitable for modern flexible logistics demands. This paper introduces a platform designed to convey and adjust cargo boxes according to their spatial positions and orientations. Additionally, a cargo pose recognition algorithm that integrates image and point cloud data are presented. By aligning depth camera data, the axis-aligned bounding box (AABB) point serves as the image’s region of interest (ROI). Peaks extracted from the image’s Hough transform are refined using RANSAC-based point cloud linear fitting, then integrated with the point cloud’s oriented bounding box (OBB). Notably, the algorithm eliminates the need for deep learning and registration, enabling its use in rectangular cargo boxes of various sizes. A comparative experiment using accelerometer sensors for pose acquisition revealed a deviation of <0.7° between the two processes. Throughout the real-time adjustments controlled by the experimental platform, cargo angles consistently remained stable. The proposed two-dimensional conveyance platform, compared to existing methods, exhibits simplicity, accurate recognition, enhanced flexibility, and wide applicability.

## 1. Introduction

Linear conveyors have become the backbone for end-to-end linear transportation tasks, primarily due to their advantages of extended conveying distances and the ability to handle substantial loads [1]. They find their major applications in the five key industries of steel, coal, cement, ports, and power, collectively occupying 85% to 90% of the conveyor belt market. With the widespread globalization of the economy and the improvement of living standards, there has been a significant surge in the demand for lightweight logistics, presenting immense potential. Notably, the evolution of online shopping has led to smaller, more diverse, and varying-sized express cargo boxes [2]. More than 80% of these boxes weigh under 15 kg, giving rise to new challenges such as sorting, consolidation, cargo box disassembly and stacking, and warehousing. Faced with these new challenges, the limitations of linear conveyors have become increasingly evident, including fixed routes, directional constraints, lack of pose adjustment, complete downtime in case of malfunction, and single-item conveyance. These shortcomings urgently require resolution.

To address the shortcomings of linear conveyors in lightweight logistics, the most primitive approach is manual assistance, which is characterized by low efficiency and high costs. Researchers have devised various automated solutions, which can be categorized into two approaches: improvement and replacement. The improvement involves upgrading the conveyor belt system. For instance, adding sorting equipment with scanning and recognition capabilities and stacking robotic arms to reduce manual labor and enhance efficiency [3,4]. However, due to the inherent structure of linear conveyors, these upgrades can only address singular issues and often lead to complex implementation processes. Furthermore, the stacking robotic arms available in the market are typically limited by linear conveyors, as they are fixed-position grabbers of a single type, capable of handling only one item at a time, resulting in substantial energy consumption and poor environmental sustainability. A novel conveyor belt system called “moveflex” has been invented, which operates without consuming electricity. It relies on gravity and multiple rolling wheels working together for transportation. While it doesn’t completely eliminate the criticisms associated with linear conveyors, it introduces a novel concept of multi-wheel collaborative cargo conveyance. Additionally, BIBA has developed a modular omnidirectional wheel platform called “celluveyor” [5], opening the era of intelligent logistics for two-dimensional omnidirectional transport within a flat surface. However, its practicality and complexity are yet to be fully realized. On the other hand, the replacement approach involves entirely forsaking the idea of improving linear conveyors and opting for alternative tools or methods to transport goods. Examples include wheeled handling vehicles with path-planning capabilities [6], guided shuttle cars [7], and material-handling robots. However, these methods often come with complex algorithms, high technical requirements, immaturity, and substantial costs.

Cargo pose recognition is the detection and determination of the position and orientation of cargo items in scenarios such as automated logistics, robotics, and industrial production. This is achieved using sensors and algorithms. Common methods for this purpose encompass computer vision, deep learning, laser radar, ultra-wideband, marker-based capture, tactile sensors, and multi-sensor fusion. Notably, computer vision stands out due to its non-contact scanning, flexible deployment, rich information extraction, real-time capabilities, and independence on external devices. The fusion of computer vision and deep learning has a wide range of applications. For instance, Hinterstoisser et al. sampled and extracted robust models for object registration, which were then subjected to model matching and ICP refinement [8,9]. Drost et al. implemented a “global modeling, local matching” strategy through descriptors, formed by a small number of point pairings on point clouds [10,11]. Spatial pose resolution can be achieved through a three-point correspondence, leading to the emergence of even more precise and robust descriptor methods [12]. Regression methods utilizing random forests establish a map linking images to model coordinates [13]. Additionally, end-to-end registration is based on neural networks [14], as well as iterative registration using probabilistic models. Although highly accurate in complex environments, these methods require substantial data and computing resources, which can limit real-time applications and decrease flexibility. Dilotis et al. recognized and located boxed cargo by segmenting planes on point clouds [15]. Building upon a successful 2D perspective image-based single-regression framework, Ali et al. extend it to generate oriented 3D object bounding boxes derived from point clouds [16]. Due to the limitations of square boxes, such as impractical spatial usage and unstable stacking, rectangular boxes are prevalent. This led to the emergence of rectangular detection algorithms based on geometric features. Two primary approaches include the Hough transform-based rectangular detection algorithm and the rectangular geometric feature-based detection algorithm [17]. Harris et al. introduce a corner feature-based rectangular detection algorithm [18], while Feng et al. propose an improved minimum bounding rectangle algorithm utilizing only point clouds [19].

This study seeks to address the drawbacks inherent in linear conveyors and the prevailing issues in cargo box pose recognition. Inspired by celluveyor, we propose a more optimized two-dimensional conveyance platform that seamlessly integrates cargo pose recognition with adjustment functionalities. Our proposed method for cargo pose recognition eliminates the complexities of extensive model training and deep learning and instead harnesses a fusion of image and point cloud data to offer adaptive and accurate recognition for rectangular cargo boxes. Additionally, we introduce an energy-efficient conveyance approach, based on distance and proximity measures to prevent unnecessary motor operation. Experimental verification was conducted to validate the recognition and adjustment of rectangular cargo boxes of varying sizes and styles.

The remainder of this paper is as follows: Section 2 elucidates the design of our two-dimensional conveyance platform. Section 3 introduces our proposed cargo box pose recognition method alongside the guiding principles of pose adjustment. Section 4 presents the experimental data acquisition using different sensors and methods under similar conditions, with subsequent comparisons to validate and analyze the adjustment process. Conclusions are then provided in Section 5.

## 2. Overall Structural Design

### 2.1. Mechanical Structure Design

The proposed two-dimensional conveyance platform employs a modular design, that offers strong reusability across production, design, and programming—effectively enhancing development efficiency. This approach inherently simplifies testing, maintenance, and upgrade phases, allowing direct access to critical components. Each independent module comprises a module controller, support plate, multiple motors, and encoders. Motor-driven wheels utilize a dual-row continuous-switching design, a specialized form of omnidirectional wheels with radiating spokes, that enable the capacity for thrust in any given direction. At the hub of the omnidirectional wheel are small passive wheels, oriented perpendicular to the primary bearing rotation direction. Devoid of independent power systems, these wheels rely on cargo friction for passive rotation, playing a pivotal role in cargo movement with minimized resistance. The support plate is divided into upper and lower tiers. The lower layer secures the module’s controller and motors, while the upper layer conceals the motors and their corresponding wiring. Strategically positioned holes in the supporting plate permit free rotation of the motor-driven omnidirectional wheels. Elevated slightly above the upper support plate, these wheels interact directly with the cargo. Their rotational movement, facilitated by friction, propels the cargo, allowing coordinated movement and adjustments across multiple modules. The combination of these independent modules forms a two-dimensional plane.

Leveraging the Intel RealSense D455 depth camera, both image and point cloud data are concurrently captured. An integrated global shutter within the RGB sensor ensures the alignment of RGB and depth information streams. This camera finds its position securely mounted, hovering above the center of the two-dimensional conveyance plane using a, held by a specialized camera mount. The entire framework of this platform is constructed using 20 × 20 profile materials, visually represented in Figure 1.

### 2.2. Distribution Design and Motion Modeling of Omnidirectional Wheels

Aiming to deliver enhanced two-dimensional conveyance and pose adjustments for various cargo box types, our study ingeniously designed the distribution of motors and omnidirectional wheels on the two-dimensional conveyance platform, ensuring its flexibility. Meticulous motion modeling, focused on the synergistic operation of omnidirectional wheel collaboration has been conducted to validate its feasibility. This research takes a pragmatic approach, streamlining control complexity and computational load, yet maintaining the foundation of two-dimensional conveyance. The overall process behind the distribution of omnidirectional wheels on this platform is depicted in Figure 2.

The process can be divided into four distinct steps:

Step 1: Two groups of omnidirectional wheels are installed orthogonally. One set is oriented horizontally, and the other vertically. When a single group rotates uniformly either forward or backward, the cargo boxes move linearly in one of four fixed directions. The intersection of the two wheel sets forms concentric circles. Within this configuration, the omnidirectional wheels align with the tangents of these circles, irrespective of their rotational speed or the direction of frictional force propelling the cargo boxes. When all four omnidirectional wheels rotate at the same speed, either clockwise or counterclockwise, they generate a unified angular velocity, creating a rotational force upon the cargo boxes.

Step 2: Remaining spaces are filled with additional omnidirectional wheels to accommodate cargo boxes of diverse dimensions. These wheels are arranged systematically to ensure that, at any given point, at least one wheel is oriented orthogonally in relation to others. This arrangement not only preserves conveyance flexibility but also facilitates cargo adjustment. This step culminates in the design completion for both cargo conveyance and adjustment areas.

Step 3: Peripheral omnidirectional wheels are added. The first row, vertically aligned, supports in ushering cargo toward the platform’s center. Flanking wheels on the left and right sides ensure conveyance flexibility, preventing cargo from falling off the platform thereby ensuring that adjustments are confined to the central area.

Step 4: The distribution design for the omnidirectional wheels is finalized and segmented into evenly sized modules. The experimental platform thus comprises 20 such modules.

#### 2.2.1. Analysis of Linear Conveyance Motion

The rotational motion of omnidirectional wheels generates tangential frictional forces upon the cargo boxes passing over them, resulting in a net force that drives the cargo in a unified direction. The specific implementation can be simplified as follows: In the first step, the translational velocity of the cargo box center is set to v_box_, with the box’s angular velocity denoted as ω. In the second step, the omnidirectional wheels’ linear velocities, v_x_ and v_y_, are calculated based on the set values. In the third step, as the cargo approaches, the omnidirectional wheels rotate at the predetermined linear velocities. Due to the concurrent presence of horizontally and vertically arranged omnidirectional wheel groups, and using motion along the X-axis as an example, the forward kinematic model when both wheel sets influence the cargo boxes can be simplified by Equation (1):(1)vboxω=v1x+v2x2v1x−v2xr1=12121r1−1r1v1xv2x

As illustrated in the linear modeling shown in Figure 3, for this case, the angular velocity ω is set to 0. When the omnidirectional wheels oriented in the X-axis direction rotate with a linear velocity of v_x_, and those in the Y-axis direction have a linear velocity of v_y_ = 0, the cargo moves along the X-axis at a velocity of v_box_. Conversely, when the omnidirectional wheels in the Y-axis direction turn at a linear velocity of v_y_ and those in the X-axis direction remain stationary with a linear velocity of v_x_ = 0, the cargo moves along the Y-axis at the same velocity, v_box_.

#### 2.2.2. Analysis of Rotational Adjustment Motion

Cargo orientation adjustments can be achieved through two distinct control methods. The first method involves the simultaneous rotation of one set of omnidirectional wheels. Here, the rotation speeds of different rows or columns are opposite, as shown in Figure 4a. According to Equation (1), in this scenario, the linear velocity of the cargo box’s center, v_box_, remains constant at 0. Consequently, the simplified formula for the cargo angular velocity can be derived from Equation (2):(2)$$\omega =\frac{v_{1}+v_{2}}{r_{1}}$$

The second method employs omnidirectional wheels arranged in a concentric circle formation with four equidistant axes at 90° angles. This facilitates a rotation adjustment of the cargo without altering its position, as illustrated in the rotational modeling in Figure 4b. The fixed-position rotation adjustment can be simplified as follows: First, determine the angular velocity ω for the cargo’s rotation. Second, using the set angular velocity, calculate the linear velocities—v_a_, v_b_, v_c_, and v_d_—of the four omnidirectional wheels that constitute the concentric circle. Finally, in scenarios where the cargo’s orientation presents a deviation angle of θ, the omnidirectional wheels commence rotation at these predetermined linear velocities. The rotation formula, based on the basic definition of angular velocity, can be expressed as given in Equation (3):(3)$$\omega =\frac{v_{a}+v_{b}+v_{c}+v_{d}}{R}=\frac{\theta }{t}$$

If the velocities of the concentric circles’ four omnidirectional wheels are equal, such that v_a_ = v_b_ = v_c_ = v_d_ = v_0_, then it can be concluded that:(4)$$t=\frac{\theta R}{4v_{0}}$$

From Equation (4), it is clear that the rotational angular velocity, ω, of the cargo boxes, the linear speed, v_0_, of the omnidirectional wheel, and the radius, R, of the concentric circle formed by the four omnidirectional wheels are fixed values. Both the angular deviation, θ, and the adjustment duration, t, have a direct proportionality. Therefore, by assessing the angular magnitude, the rotation-adjustment time can be ascertained, corresponding to a set PWM waveform output time for driving the motors.

### 2.3. Control Unit

In terms of operation, the control module utilizes a master-slave control mode. The main control unit is responsible for receiving information from the upper computer, performing the necessary calculations, and sending commands to the respective module control units. These module control units, upon receiving instructions from the main control unit, control the rotation of motors to accomplish either cargo conveyance or pose adjustments. To guarantee prompt reactions during the control phase, a high-performance microcontroller, boasting a maximum clock frequency of 72 MHz was employed as the control chip, ensuring real-time response in the control process of this study. Group control of the two-dimensional convey platform is achieved through controller area network (CAN) bus control. The motor driver employed is the A4950, which is an integrated full-bridge circuit motor driver chipset. This chip receives commands from the control chip, sending pulse width modulation (PWM) signals to the motors. It also receives encoder feedback, using it to control motor speed through proportional-integral (PI) control, ensuring steady operation. Due to the limited number of timers available in the microcontroller and the need to address the control requirements of nine motors within a single module simultaneously, this research employs two distinct analog switch chips. These are applied to the encoder’s input and the PWM’s output terminals, respectively, effectively expanding the communication pathways to accommodate the demands of multi-channel control. Figure 5 shows the connection diagram and circuit diagram of the single module control unit of the platform.

## 3. Pose Recognition and Adjustment

### 3.1. Estimation of Cargo Box Angular Deviation from Image Acquisition

The orientation deviation of the cargo can be computed by extracting features associated with its alignment. In this study, the cargo’s location is designated as the “region of interest” (ROI). The Canny edge detection technique is applied to the ROI, resulting in a binary image. Utilizing the Hough transform, the Cartesian coordinate lines represented by the equation y = kx + b within this binary image undergo a transformation into a parametric equation, as shown in Equation (5). This equates to the parameterization of lines in the context of the polar-coordinate parameter space:(5)$$r=x*\cos\left(\theta \right)+y*sin(\theta )$$

When straight lines in the Cartesian coordinate system are mapped to the polar-coordinate parameter space, they correspond to intersections. By extracting the peaks from these intersections, the longest straight-line segments are identified. From these segments, both the gradient and the angle of inclination can be directly calculated. The inclination angle, θ, is then determined by solving Equation (6).
(6)$$\theta =\frac{\tan^{-1}\frac{\left(y_{2}-y_{1}\right)}{\left(x_{2}-x_{1}\right)}*180}{\pi }$$

Figure 6 visualizes the methodology to acquire the orientation angles of the cargo box from the captured images. Within this process, the two highest peaks are extracted, and subsequently, the slope and angle of the corresponding lines are calculated.

While images consist of an arrangement of pixels in a two-dimensional matrix, offering higher information density, ensuring accurate feature extraction, and rapid processing, they lack in providing spatial information regarding the positioning of the cargo. This is a significant drawback in recognition tasks. Moreover, features extracted from these images might include unwanted noise points, which, instead of aiding, act as disturbances in the recognition process. Such discrepancies necessitate further refinement and selection.

### 3.2. Point Cloud-Based Localization and Pose Estimation of Cargo Boxes

Point clouds provide a wealth of three-dimensional spatial information, with each scan point encapsulating its own set of three-dimensional coordinates. Due to the relatively fixed positions of the two-dimensional convey platform and depth camera, deducing the three-dimensional coordinates of the two-dimensional convey platform in relation to the depth camera’s coordinate system remains constant. This consistency was harnessed to adopt an attribute-based filtering method, focusing on attributes such as coordinate positions, to extract the point cloud data of the cargo box. Alternatively, fixed geometric spatial constraints can be imposed to achieve targeted control objectives. The process begins by determining the three-dimensional coordinates of the two-dimensional convey platform, its modules, and omnidirectional wheels within the depth camera’s coordinate system. Techniques such as point cloud conditional filtering and radius-based filtering then define attribute ranges. This process effectively removes extraneous elements such as the ground, platform, and camera support, concentrating solely on the point cloud data of cargo boxes positioned above the platform. Figure 7 details the extraction process for the cargo box point cloud.

After extracting the top surface of the cargo box point cloud, its edges were delineated. The random sample consensus (RANSAC) [20] algorithm was employed on the extracted top-surface point cloud, allowing for the recognition of rectangular edge lines. This procedure aids in determining the cargo box’s angular misalignment. RANSAC operates by iteratively selecting random samples from the point cloud dataset, classifying samples consistent with the model as “inliers” and inconsistent ones as “outliers”. This process is continued until a model that fits most closely within a predefined range is obtained, typically representing the longest line segment within the point cloud edges. For RANSAC’s sampling iterations involving n points in the point cloud and boasting a confidence level of P (95–99%), the termination iteration count, h, can be obtained using Equation (7):(7)$$h\geq \frac{\log\left(1-P\right)}{\log\left(1-{\left(\frac{inliers\;number}{inliers\;number+outliers\;number}\right)}^{n}\right)}$$

Within this study, each iteration yields the longest edge in the point cloud, and subsequent calculations deal with the remaining points. This iterative approach continued until all four edge lines of the rectangular cargo box point cloud were ascertained. Figure 8 depicts the arbitrary edge lines drawn from the point cloud via RANSAC’s iterative fitting mechanism.

The point cloud bounding box algorithm computes bounding boxes for point cloud data, providing valuable geometric information for point cloud processing tasks. In this study, we utilized both the axis-aligned bounding box (AABB) and the oriented bounding box (OBB) techniques from the point cloud bounding box algorithm. The AABB is determined by directly traversing the point cloud to find the extremities of the coordinate points, forming an approximation of the point cloud boundary as a box, with all six faces parallel to the coordinate axes. In contrast, the OBB generates a bounding box that can rotate relative to these axes, adapting its size and orientation based on changes in the point cloud’s size and orientation. To compute the point cloud normal vectors and construct the OBB, we utilized principal component analysis (PCA) [21]. Assuming each point p_i_ in the point cloud has a neighboring set {p_1_, p_2_, …, p_k_}, where k represents the number of nearest neighbors and p_i_’s coordinates are (x_i_, y_i_, z_i_), the point cloud’s covariance matrix C is calculated using Equation (8):(8)$$C=\frac{1}{k}\sum_{i=1}^{k}(p_{i}-m){\left(p_{i}-m\right)}^{T}$$
where m represents the average position of the K-nearest neighbors within the point cloud:(9)$$m=\frac{1}{k}\sum_{i=1}^{k}p_{i}$$

By distributing points evenly across the coordinate axes and leveraging linear algebra principles, the covariance matrix is diagonalized: |A − λE| = 0 facilitates the extraction of eigenvalues λ_1_, λ_2_, λ_3_. Through back-substitution, we acquire the corresponding eigenvectors ε_1_, ε_2_, ε_3_. The primary OBB axis direction aligns with the eigenvector corresponding to the largest eigenvalue. As depicted in Figure 9, PCA was utilized to approximate the point cloud normal vectors and design the point cloud OBB. The lower-left corner of Figure 9 displays the pertinent output information related to the point cloud’s OBB.

In comparison to imagery, point cloud data demonstrate shape invariance during rigid transformations, implying the robustness of point cloud processing algorithms against object rotations and translations. However, dealing with large point cloud datasets poses computational challenges, while small datasets can lead to inaccurate calculations. Consequently, even if the position of bounding boxes is accurate, determining the main orientation of different-sized cargo boxes can fluctuate in accuracy. Point clouds, being composed of discrete points, are susceptible to noise interference, particularly along edges where stability is often compromised during application. This susceptibility intensifies in cases with varying noise levels, leading to pronounced changes and instability areas along the boundaries. RANSAC’s suitability for fitting edge lines in point clouds allows for stable extraction of these edges. However, the calculated values were not consistently stable, exhibiting real-time variations and potential errors when discerning the longer from the shorter edges.

### 3.3. Pose Recognition by Fusing Image and Point Cloud Data

By separately investigating cargo box pose recognition using image and point cloud approaches, the distinct advantages and disadvantages of each method become evident. This research integrates image and point cloud data to overcome the shortcomings inherent in each method. Firstly, alignment of the image with the point cloud data is achieved. A variety of point cloud filtering techniques are employed to extract compliant top-surface data for the cargo boxes. Using the point cloud AABB, the box’s position in relation to the depth camera’s coordinate system is discerned. This position then utilizes the ROI within the image, where the Hough transform is then applied to extract line segments. Edges were derived from the filtered point cloud data, followed by the implementation of RANSAC for line fitting. Comparing the point cloud edge lines with those from image processing verified the alignment of image-derived lines with the top edges of cargo boxes, refining the image-based Hough line detection. Post-refinement, the longest line, representative of the box’s longer edge, was identified. The derived inclination angle signified the cargo box’s angular deviation. Figure 10 presents comparative experiments conducted on varied-sized and styled rectangular boxes. The upper portion shows the extraction of the two highest peak points from image processing, forming the basis for slope and angle calculations. The lower section contrasts the results from peak point extraction and RANSAC line fitting post-validation. This comparison highlights that combining image-based Hough line extraction with RANSAC point cloud line fitting consistently extracts the longer top edge of cargo boxes, thereby obtaining the overall angular deviation of the box.

Upon jointly applying point cloud edge fitting and the image-based Hough transform, the angle of the chosen longest line is calculated. This angle is then compared and fused with the angle derived from the PCA of the point cloud’s OBB main direction. In cases of similar angles, an averaged value was taken. However, if a significant difference is detected, the angle obtained from image data is adopted. In cases where image-based edge angle acquisition proves unattainable, the angle obtained from the PCA’s main OBB direction is employed. Figure 11 illustrates this comprehensive operational paradigm.

### 3.4. Implementation of Cargo Pose Adjustment through Distance and Proximity

In this study, the microcontroller supports low-power modes such as sleep and standby. These modes allow real-time comparison between independent modules and cargo box positions to determine whether a module should be operational or on standby. With the fixed positioning of the two-dimensional conveyor platform and camera, the platform’s position relative to the camera’s coordinate system remains constant. If point cloud data are present at the platform’s outermost omnidirectional wheels, it indicates that cargo is either entering the platform or is at risk of falling. In these cases, no calculations are performed. The outermost omnidirectional wheel rotates inward, guiding the cargo to the platform’s central conveyor and adjustment area. When point cloud data are exclusively situated within this central region, it implies that the cargo has completely entered the platform, triggering pose calculation data processing. On the two-dimensional plane, each independent module corresponds to a fixed center point coordinate (x_n_, y_n_) (where n = 1, 2, 3, …) within the camera’s coordinate system. Each of these coordinates is enclosed within a circle, centered on the module’s center point and with a radius of r_1_. Regardless of its orientation, the cargo box’s body remains within a circle with center (X, Y) and radius r_2_. The upper-level computer sends real-time cargo box information from the depth camera to the two-dimensional conveyor platform’s primary control unit through a serial port. This main control unit then conducts instantaneous calculations and comparisons between the pre-stored center point coordinates (x_n_, y_n_) (where n = 1, 2, 3, …) of each module and the cargo box information. This relationship is expressed by Equation (10):(10)$$d_{n}=\sqrt{{\left(X-x_{n}\right)}^{2}+{\left(Y-y_{n}\right)}^{2}}\;\;(n=1,2,\ldots )$$
where d_n_ represents the relative distance between each module and the cargo. By comparing it with the combined radius, r (r = r_1_ + r_2_), the module currently positioned on the cargo box can be determined. This enables the identification of modules that need to be operational and those that can remain in standby mode, thereby preventing wasteful motor rotations and conserving resources. The underlying principle and methodology are depicted in Figure 12.

This study’s upper-level computer system employs the robotics operating system (ROS) [22] on the Linux platform, capitalizing on both the OpenCV and PCL libraries for its processing. Outcomes were consolidated through the nodes’ publication and subscription, and these amalgamated data were communicated to the main control unit for further processing using serial communication. As shown in Figure 13, it illustrates the synchronization of timestamps between rviz simulation and ROS nodes. The upper-level computer sends the processed angle data to the main control unit, which, after a one-second delay, returns the information unchanged to validate successful communication.

## 4. Results

### 4.1. Experimental Data

An industrial-grade accelerometer sensor was employed in this study to validate the experimental data. The accelerometer sensor was calibrated, achieving an accuracy of within 0.2° along the Z-axis (horizontal angle axis). The accelerometer sensor was securely mounted on the experimental cargo box and rotated along with the box’s movement. Figure 14 illustrates the setup and experimental environments.

The experimental data collection underwent nine adjustments, each made from the initial pose. These adjustments were approximately 10° apart, determined by observing the Z-axis values from the accelerometer sensor while rotating the experimental cargo box. The detailed adjustment process is illustrated in Figure 15.

After each adjustment, four angle datasets were recorded: the accelerometer sensor’s angle (control group), the angle obtained through mutual filtering of point cloud and image datasets, the angle from the point cloud’s OBB, and the angle following the complementary fusion of the image and point cloud datasets. The recorded angle values are presented in Table 1.

The specific orientation angle of the box is influenced by variation due to reference points and various other factors. Therefore, this study undertook a comparative analysis of the same cargo box, under equivalent conditions, using different methods to obtain the angles. The differences in angles before and after each adjustment from Table 1, as well as the overall changes, were used to create Table 2.

A comparison of the four datasets revealed that, under the same experimental conditions, the image-based angle estimation exhibits higher accuracy. Although the discrete point cloud data are stable, it is also susceptible to noise, which can affect its precision. However, these data still provide a reference. By fusing information from both image and point cloud sources, the accuracy can be further improved, reducing errors during the variation process.

Testing outcomes indicated that precise cargo box orientation angles can be achieved without employing deep learning or registration techniques, by solely processing data from depth cameras. This approach proves effective for adjusting cargo box poses on a two-dimensional conveyor platform. The system’s capability to retrieve cargo pose and make real-time adjustments was verified by interfacing with the main control chip through serial communication. When cargo boxes of arbitrary orientations were placed in the two-dimensional conveyor platform’s adjustment area, the depth camera recognized the boxes and adjusted their angles. Table 3 illustrates the real-time angular adjustments, starting from an initial angle of 71.24° and adjusting to a near-horizontal position of approximately 0°.

The experimental platform utilizes omnidirectional wheels with a diameter of 58 mm and a set of fixed omnidirectional wheels arranged in concentric circles with a radius R of 0.1 m. The cargo box adjustment experiment was conducted based on a foundation where the line speed v_0_ of the omnidirectional wheels was set at 120 revolutions per minute. The adjustment was completed within 4–5 s, aligning with the rotational adjustment motion analysis outlined in Equation (4). To mitigate the effects of inertia, we set a stop criterion: the adjustment operation ceased when the identified angular deviation from the expected result was <1°. In other words, the adjustment operation was halted after recognizing an angle of 0.962°, and the final angle at the point of cessation was 0.138°, with minimal deviation from the expected value of 0°, indicating a high level of precision.

### 4.2. Experimental Results Comparison

We initialized all four angle estimation methods from a base value of 0 and then made adjustments using incremental changes. The resulting comparative data are shown in the bar chart given in Figure 16. This comparison revealed that angles obtained from the complementary fusion of image and point cloud information closely match those acquired by the accelerometer sensor, with only a maximum difference of 0.7° and a total variation difference of merely 1.24°. Even with the least performing method, the point cloud OBB angle, the total variation difference remained modest at 3.3°. This approach, which avoids the need for deep learning or model registration, offers low complexity, minimal computational requirements, and wide applicability. It proves to be effective in real-world applications for cargo pose adjustments.

Figure 17 presents a comparison between the real-time angular adjustments of cargo boxes on the two-dimensional convey platform from Table 3 and the ideal trend line. The ideal trend line represents the most optimal trend of angular adjustment in the absence of interference and errors. By comparison, it can be observed that during the actual correction process on the two-dimensional convey platform, the deviation angle of the cargo boxes changes relatively smoothly, allowing for the adjustment of the cargo box’s orientation to meet the predetermined requirements, with a final error of less than 1.5°.

## 5. Conclusions

In our study, we investigated a two-dimensional conveyance system based on deep camera pose recognition. This platform can be used in conjunction with linear conveyance devices, serving as an aid and decision-making tool during various phases of logistics operations. It also possesses the capability to independently perform multi-directional conveyance, recognition, and pose adjustments. This contributes to the optimization of cargo box logistics, reduction in energy consumption, and the enhancement of automation and efficiency. We introduced a recognition method that fuses image and point cloud information. By leveraging the advantages of pixel-level processing in images and position-based processing in point clouds, we bypassed the complex procedures associated with deep learning and registration. Instead, the methodology directly targets the precise identification of the rectangular box’s edge features to determine and compute its posture. The system’s physical development and the fundamental control of the two-dimensional conveyance system were brought to realization. Experimentation and verification were conducted to retrieve the cargo pose, emphasizing determining the posture angles with precision. This led to real-time and accurate readings, facilitating precise communication between the top-tier computer and the main control unit via the ROS. The conveyance approach presented in this study exhibits increased intelligence, flexibility, and energy efficiency. The posture-recognition method’s complexity is low, it showcases wide adaptability, and its precision and real-world applicability are commendable.

## Figures and Tables

**Figure 1 sensors-23-08754-f001:**
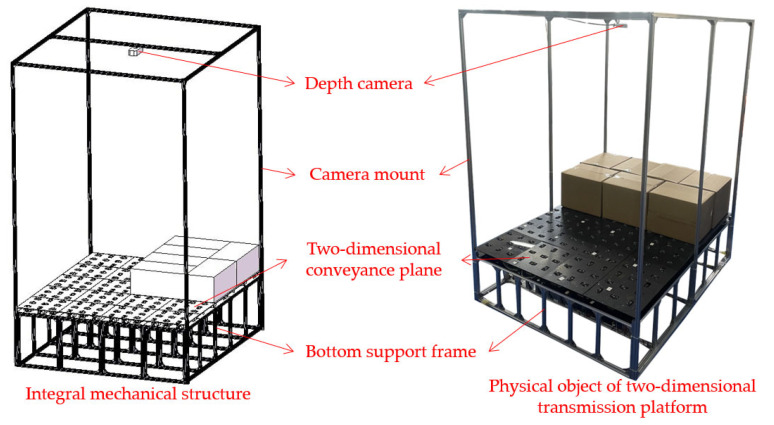
Overall structure of two-dimensional conveyance platform.

**Figure 2 sensors-23-08754-f002:**
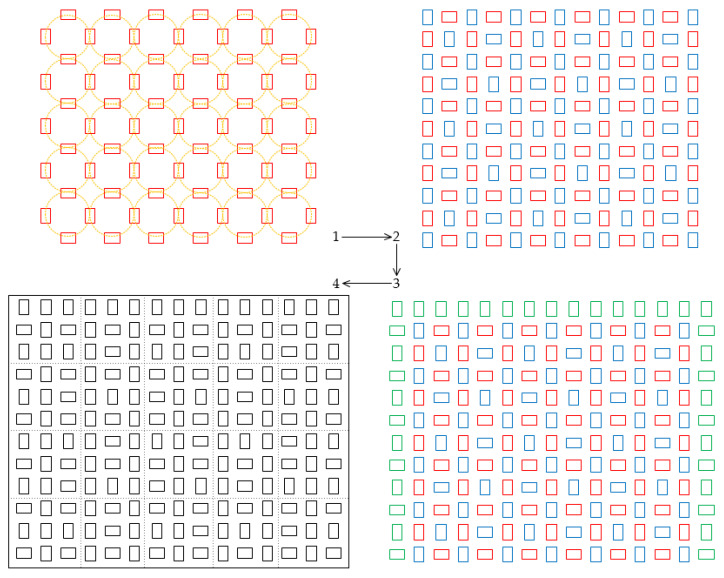
Flowchart of omnidirectional wheel distribution design.

**Figure 3 sensors-23-08754-f003:**
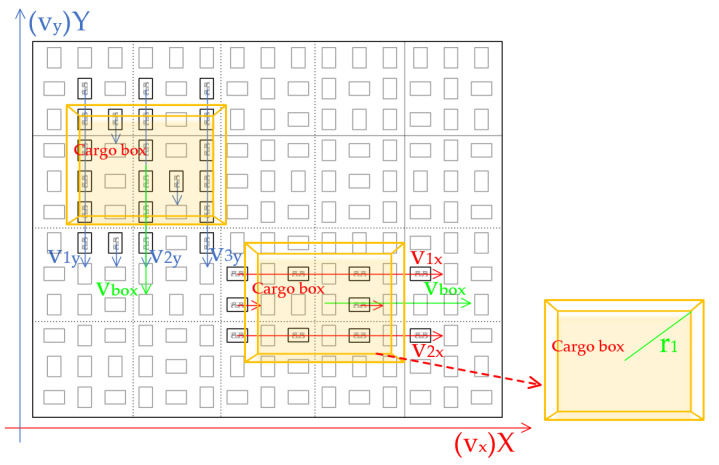
Modeling of single-module linear conveyance motion.

**Figure 4 sensors-23-08754-f004:**
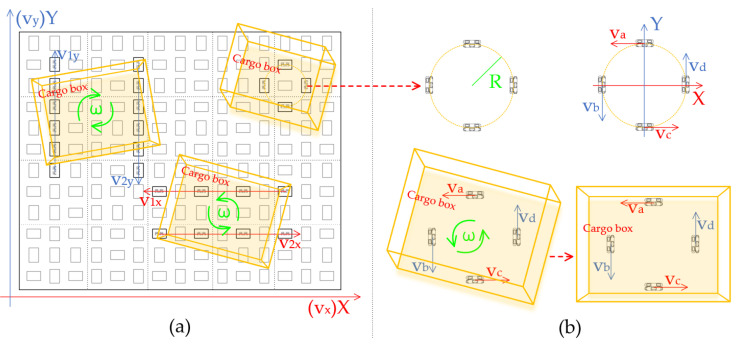
Modeling of rotational adjustment motion. (**a**) Modeling the movement of cargo boxes adjusted by reverse rotation between different rows or columns of the same group of omnidirectional wheels; (**b**) Modeling of cargo box motion with fixed-position concentric omnidirectional wheel adjustment.

**Figure 5 sensors-23-08754-f005:**
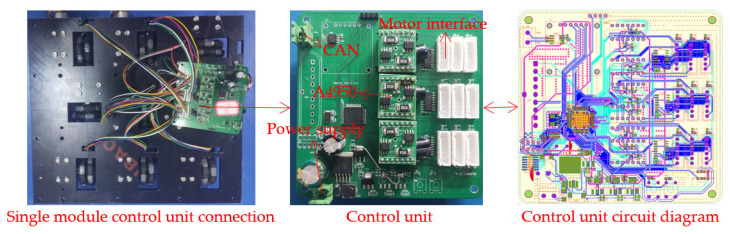
Circuit diagram of a single module’s control unit.

**Figure 6 sensors-23-08754-f006:**
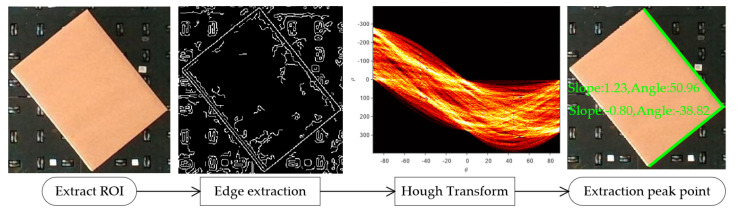
Processing workflow for obtaining cargo box attitude angles from images.

**Figure 7 sensors-23-08754-f007:**
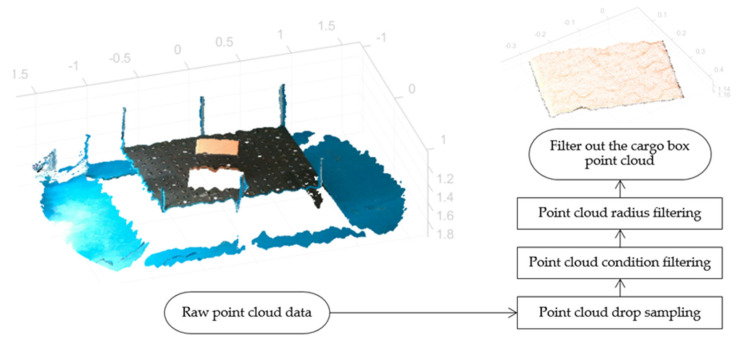
Workflow for cargo box point cloud extraction.

**Figure 8 sensors-23-08754-f008:**
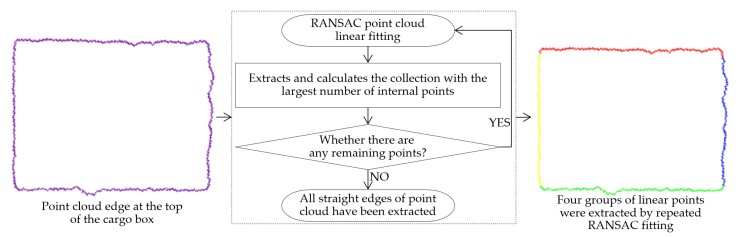
Repetitive RANSAC fitting for extracting arbitrary line edges from point clouds.

**Figure 9 sensors-23-08754-f009:**
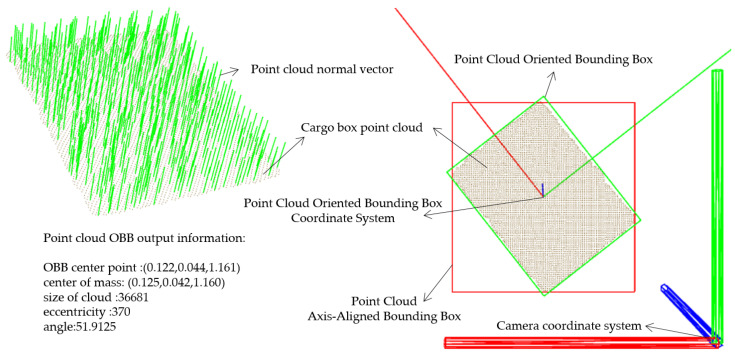
Point cloud normal vector and point cloud bounding box.

**Figure 10 sensors-23-08754-f010:**
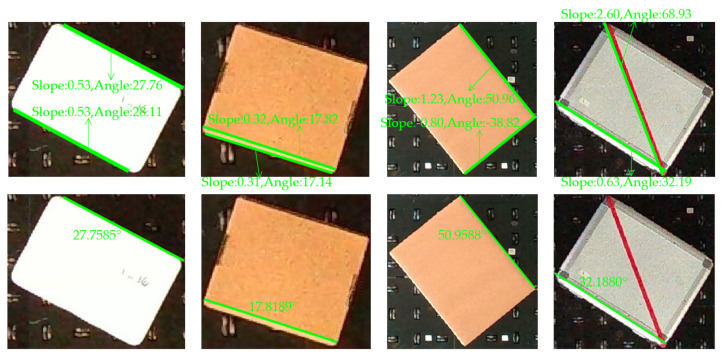
Comparison and selection between image Hough line extraction and RANSAC point cloud edge line fitting.

**Figure 11 sensors-23-08754-f011:**
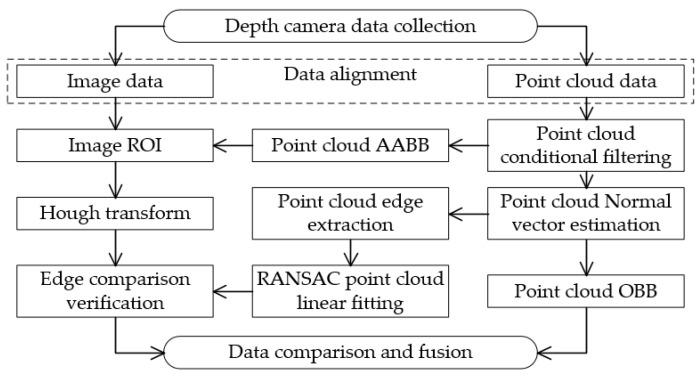
Fusion principle of image and point cloud.

**Figure 12 sensors-23-08754-f012:**
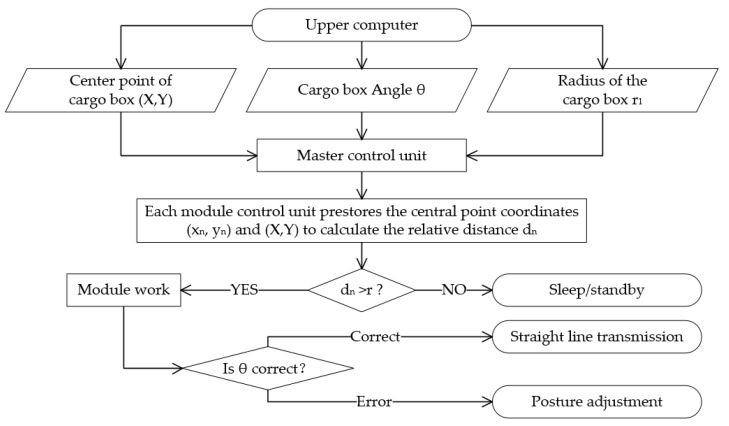
Principle and workflow of pose adjustment implementation.

**Figure 13 sensors-23-08754-f013:**
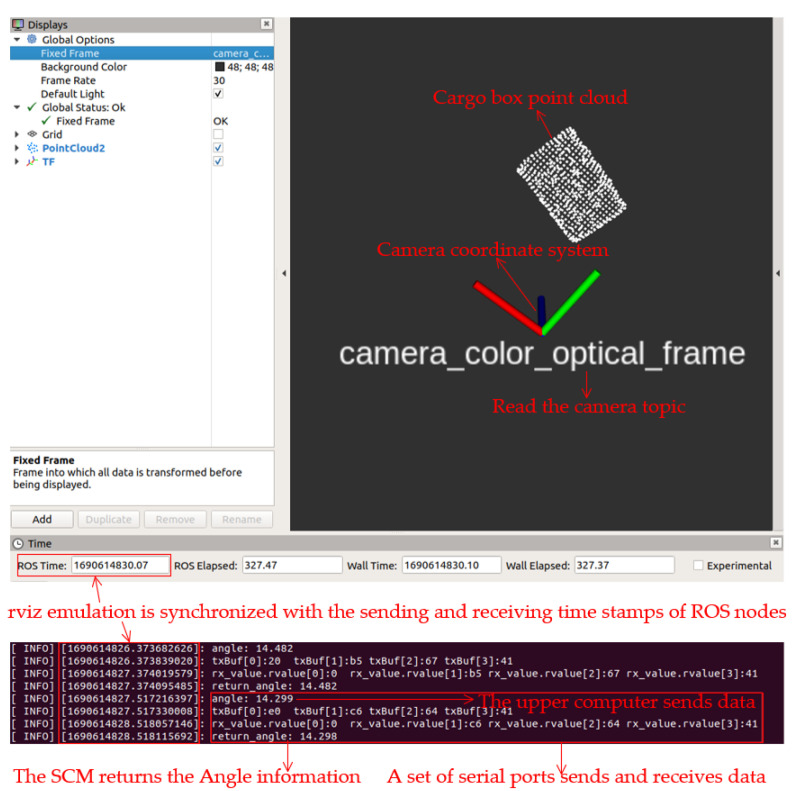
Synchronization of timestamps between rviz simulation and ROS nodes.

**Figure 14 sensors-23-08754-f014:**
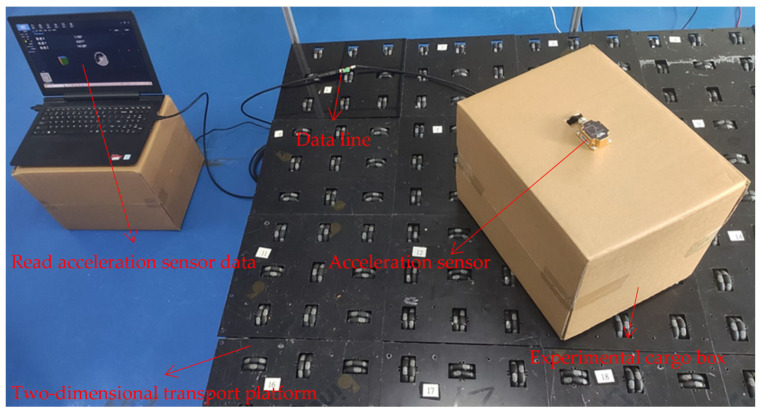
Installation of accelerometer sensor and experimental environment.

**Figure 15 sensors-23-08754-f015:**
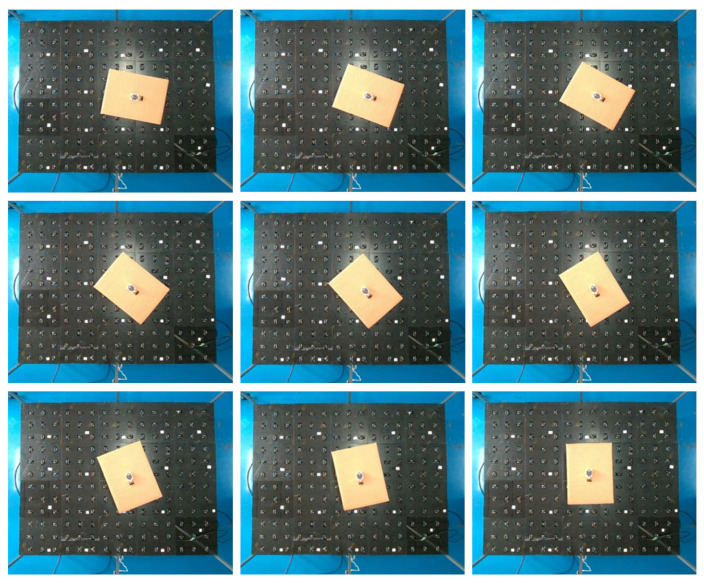
Process of box rotation variation.

**Figure 16 sensors-23-08754-f016:**
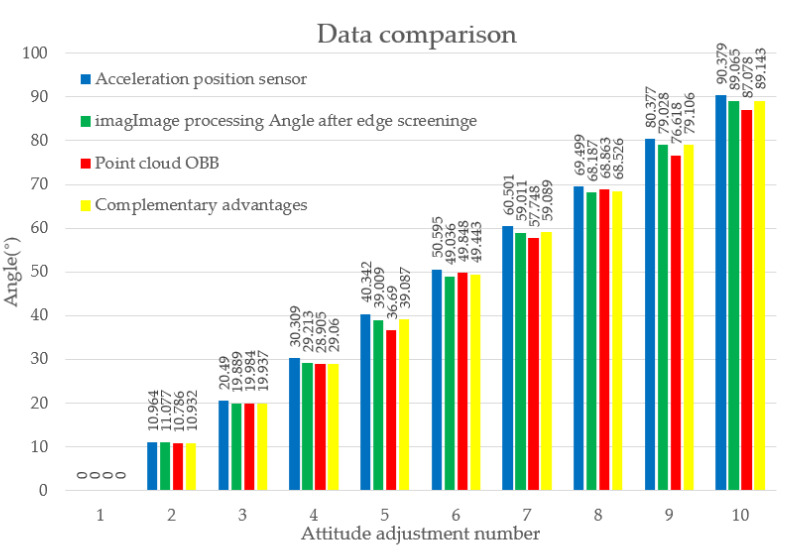
Comparative analysis of angle estimation methods.

**Figure 17 sensors-23-08754-f017:**
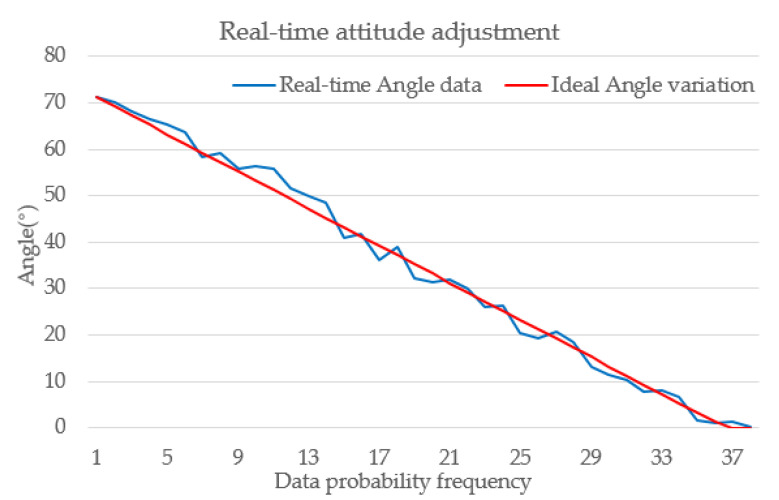
Real-time variation of cargo box angles during adjustment process and ideal trend line.

**Table 1 sensors-23-08754-t001:** Data acquisition during experimental cargo box adjustment.

Number of Adjustments	Control Group (°)	Image (°)	Point Cloud OBB (°)	Data Fusion (°)
0	159.292	−1.042	−1.197	−1.120
1	148.328	10.035	9.589	9.812
2	138.802	18.847	18.787	18.817
3	128.983	28.171	27.708	27.940
4	118.950	37.967	35.493	37.967
5	108.697	47.994	48.651	48.323
6	98.791	57.969	56.551	57.969
7	89.793	67.145	67.666	67.406
8	78.915	77.986	75.421	77.986
9	68.913	88.023	85.881	88.023

**Table 2 sensors-23-08754-t002:** Angle variation before and after adjustment.

Number of Adjustments	Control Group (°)	Image (°)	Point Cloud OBB (°)	Data Fusion (°)
0	0	0	0	0
1	10.964	11.077	10.786	10.932
2	9.526	8.812	9.198	9.005
3	9.819	9.324	8.921	9.123
4	10.033	9.796	7.785	10.027
5	10.253	10.027	13.158	10.356
6	9.906	9.975	7.785	9.646
7	8.998	9.176	11.115	9.437
8	10.878	10.841	7.755	10.580
9	10.002	10.037	10.460	10.037
Total variation	90.379	89.065	87.078	89.143

**Table 3 sensors-23-08754-t003:** Real-time angle changes during cargo box adjustment.

Record Times	Angle (°)	Record Times	Angle (°)	Record Times	Angle (°)
1	71.240	14	48.537	27	20.817
2	70.078	15	40.900	28	18.534
3	68.074	16	41.889	29	13.122
4	66.512	17	36.145	30	11.486
5	65.402	18	38.920	31	10.181
6	63.588	19	32.261	32	7.746
7	58.223	20	31.449	33	8.198
8	59.304	21	32.030	34	6.598
9	55.823	22	29.885	35	1.720
10	56.258	23	26.057	36	0.962
11	55.751	24	26.211	37	1.445
12	51.553	25	20.394	38	0.138
13	49.892	26	19.272		

## Data Availability

No new dataset created.

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
