# Peer review of "Design of a Two-Dimensional Conveyor Platform with Cargo Pose Recognition and Adjustment Capabilities"

_sensors, 2023, doi:10.3390/s23218754_

Round 1
Reviewer 1 Report
This topic is interesting, but there are still some issues. We suggest that it can be accepted after major repairs.
1. The author did not provide further research results on linear transmission machines and a comparison of the advantages and disadvantages of related algorithms in the introduction section. This is the core of their research, and the innovation points are not prominent. To our knowledge, linear transmission machines have become very mature and have been widely used in logistics transfer, docks, airports, stations, and other places. We suggest that the author conduct thorough survey and provide detailed comparisons in the introduction section, preferably with data support, in order to be convincing and increase readers' interest, otherwise it will be meaningless.
2. The explanation of some of the figures in the article is not in place, such as Figure 16. The author did not provide a detailed explanation of the figures and did not provide a conclusion. Some other images also have similar issues, please make modifications.
3. The author should provide the speed and accuracy of goods operation on the platform, which is also related to whether this platform is meaningful.
This topic is interesting, but there are still some issues. We suggest that it can be accepted after major repairs.
1. The author did not provide further research results on linear transmission machines and a comparison of the advantages and disadvantages of related algorithms in the introduction section. This is the core of their research, and the innovation points are not prominent. To our knowledge, linear transmission machines have become very mature and have been widely used in logistics transfer, docks, airports, stations, and other places. We suggest that the author conduct thorough survey and provide detailed comparisons in the introduction section, preferably with data support, in order to be convincing and increase readers' interest, otherwise it will be meaningless.
2. The explanation of some of the figures in the article is not in place, such as Figure 16. The author did not provide a detailed explanation of the figures and did not provide a conclusion. Some other images also have similar issues, please make modifications.
3. The author should provide the speed and accuracy of goods operation on the platform, which is also related to whether this platform is meaningful.
Reviewer 2 Report
1. While the methods section provides a good overview, offering more detailed explanations or visual aids might help in enhancing clarity and understanding.
2. The paper's presentation can be improved by ensuring a consistent flow of information and possibly including more illustrative figures or diagrams where complex concepts are introduced.
3. Consider expanding on the real-world applicability and potential impact of the research in the conclusion, emphasizing its relevance in practical scenarios.
Reviewer 3 Report
Congratulations to the authors of a good article. In their research, they proposed the design of a two-dimensional loaded conveyor platform and the ability to recognise and adjust position
The article was written to a high standard.
I have one comment: they propose to extend the literature review.
Round 2
Reviewer 1 Report
The author has completed all modifications and suggests that this manuscript be accepted.